# Artificial Intelligence and Machine Learning in the Diagnosis and Management of Gastroenteropancreatic Neuroendocrine Neoplasms—A Scoping Review

**DOI:** 10.3390/diagnostics12040874

**Published:** 2022-03-31

**Authors:** Athanasios G. Pantelis, Panagiota A. Panagopoulou, Dimitris P. Lapatsanis

**Affiliations:** 14th Department of Surgery, Evaggelismos General Hospital of Athens, 10676 Athens, Greece; dimitrislapatsanis@gmail.com; 2Protypo Dialysis Center of Piraeus, 18233 Piraeus, Greece; giota81@gmail.com

**Keywords:** neuroendocrine tumors, neuroendocrine neoplasms, carcinoid, gastroenteropancreatic, GEP-NETs, Pan-NENs, SI-NETS, artificial intelligence, machine learning, deep learning

## Abstract

Neuroendocrine neoplasms (NENs) and tumors (NETs) are rare neoplasms that may affect any part of the gastrointestinal system. In this scoping review, we attempt to map existing evidence on the role of artificial intelligence, machine learning and deep learning in the diagnosis and management of NENs of the gastrointestinal system. After implementation of inclusion and exclusion criteria, we retrieved 44 studies with 53 outcome analyses. We then classified the papers according to the type of studied NET (26 Pan-NETs, 59.1%; 3 metastatic liver NETs (6.8%), 2 small intestinal NETs, 4.5%; colorectal, rectal, non-specified gastroenteropancreatic and non-specified gastrointestinal NETs had from 1 study each, 2.3%). The most frequently used AI algorithms were Supporting Vector Classification/Machine (14 analyses, 29.8%), Convolutional Neural Network and Random Forest (10 analyses each, 21.3%), Random Forest (9 analyses, 19.1%), Logistic Regression (8 analyses, 17.0%), and Decision Tree (6 analyses, 12.8%). There was high heterogeneity on the description of the prediction model, structure of datasets, and performance metrics, whereas the majority of studies did not report any external validation set. Future studies should aim at incorporating a uniform structure in accordance with existing guidelines for purposes of reproducibility and research quality, which are prerequisites for integration into clinical practice.

## 1. Introduction

Neuroendocrine neoplasms (NENs) of the gastrointestinal tract and the pancreas are rare tumors that tend to be diagnosed incidentally but with an increasing frequency [1,2]. GEP-NENs arise from the neural crest and may be located in the stomach, the small intestine, the appendix, the colon, the rectum, the pancreas, the ampulla of Vater, and the extrahepatic bile ducts, as well as the liver in the form of metastases. For the purposes of this review, we will focus on the former group of organs. For purposes of systematization, NENs can be divided into well differentiated neuroendocrine tumors (NETs) and poorly differentiated neuroendocrine carcinomas (NECs), the latter representing 10–20% of NENs [3]. This classification is not arbitrary, as NETs and NECs represent two genetically and biologically separate entities. NETs may be further classified into NETs arising from the gastrointestinal tract (GI-NETs, also known as carcinoids; ~50% of GEP-NETs) and ones affecting the pancreas (Pan-NENs; ~30% of GEP-NETs). NENs may or may not be functional. Nonfunctioning NENs are usually asymptomatic (especially early-stage ones), but may cause gastrointestinal bleeding and anemia, as well as obstructive effects which may present as jaundice, small bowel obstruction, intussusception, appendicitis and palpable abdominal mass depending on their anatomic location. Functioning GI-NENs may cause flushing, diarrhea, endocardial fibrosis and wheezing, owing to the synergistic effect of secreted vasoactive substances such as prostaglandins, kinins, serotonin and histamine. These symptoms signal the so-called carcinoid syndrome and usually herald liver metastases, because normally the liver inactivates products secreted into the portal circulation [4]. On the other hand, functioning Pan-NENs cause distinctive syndromes depending on the secreted product (i.e., gastrinoma–Zollinger-Ellison syndrome (ZES), insulinoma–Whipple’s triad, glucagonoma–necrolytic erythema and hyperglycemia, VIPoma–watery diarrhea-hypokalemia-achlorhydria syndrome, somatostatinoma–diabetes, gallstone formation and steatorrhea etc) [1,2]. Gastric NETs merit special mention, as they may manifest with atypical symptoms that are not related to hormone secretion [1]. Type 1 gastric NETs (70–80% of gastric NETs) are related to atrophic gastritis that leads to secondary hypergastrinemia, which in turn causes hyperplasia of the enterochromaffin-like (ECL) cells. With continuous stimulation, ECLs give rise to aggregates which constitute foci of NETs. Type 2 gastric NETs (approximately 30%) are associated with ZES and multiple endocrine neoplasia type 1 (MEN-1). Type 3 gastric NETs are not related to other syndromes, are sporadic and are the most aggressive, as they tend to metastasize in 50–100% of the cases. Finally, type 4 gastric NETs are poorly differentiated and typically non-amenable to surgical manipulations.

Various biomarkers (mainly in immunohistochemistry) serve different purposes in the spectrum of NENs: Ki-67 is the most well-known among them, it has a prognostic relevance and is an essential component of the WHO grading of NENs [5]; SSTR-2/5 are useful for the detection of somatostatin receptors when functional imaging (with ^68^Ga-DOTATATE PET/CT) is not possible; DAXX/ATRX has a prognostic relevance for Pan-NETs and is useful for distinguishing between NETs and NECs; p53/pRb are used for the classification of poorly differentiated NECs and the distinction from G3 NETs; and MGMT has a predictive response for the chemotherapeutic temozolomide [3]. Chromogranin A (CgA) is a useful circulating biomarker, especially for the diagnosis of asymptomatic NETs [1]. The NETest is a multigene mRNA assay that provides a broad molecular characterization GEP-NENs with high sensitivity and specificity and better diagnostic accuracy when compared to isolated biomarkers such as CgA [2]. Functional imaging with ^68^Ga-DOTATATE, which binds to somatostatin receptors (SSRTs), is the cornerstone of diagnosis (and particularly localization and staging) of NETs, especially in the cases of small intestinal NETs (SI-NETs), large NETs and metastatic NETs [1].

Artificial intelligence (AI) is the process of simulating human learning by a machine, in the context of which large quantities of digitized data (input) are fed to a computer, the computer processes them with the aid of AI algorithms, and it ultimately reaches conclusions, makes decisions, or adjusts its function (output). Input data may derive from electronic health records (EHRs) and large databases, such as the Surveillance, Epidemiology, and End-Results Program (SEER) registry, digitized histology samples and whole slide images (WSIs), digital imaging studies (computed tomography—CT, magnetic resonance imaging—MRI, endoscopic ultrasonography—EUS, positron emission tomography—PET etc.), endoscopic study videos and so forth. AI is an umbrella term and includes supervised machine learning (ML), unsupervised machine learning, deep learning (DL) and reinforcement learning [6]. Each discipline differs from the preceding one in that it entails a greater degree of autonomy from the operator’s supervision. AI with its subcategories is gradually entering healthcare and pertinent studies have had an exponential publication rate over the last five years, with various applications being integrated into clinical practice [7]. For the non-familiar clinician, AI should not be deemed as a substitute to their pivotal role in the patient care continuum or as an incomprehensible field belonging exclusively to computer experts but should rather be approached as a valuable tool in the process of decision-making, as well as a novel statistical method which, unlike traditional ones, may reveal hidden relationships between causes of disease and diagnosis, management and potentially cure.

With the present study we attempt to map the current status of AI and its applications in the diagnosis and management of gastroenteropancreatic NENs (GEP-NENs). Given on the one hand that NENs are relatively rare entities and on the other hand that AI, ML and DL are novel in the field of Medicine, we deemed it a rather uncharted area of interest and opted for a scoping review.

## 2. Materials and Methods

This review was performed according to the PRISMA extension for scoping reviews [8]. We performed literature search using the PubMed database in January 2021. The combined search terms were [artificial intelligence; machine learning; deep learning] AND [neuroendocrine; NET; NEN; carcinoid; insulinoma; glucagonoma; gastrinoma; VIPoma] AND [gastrointest*; GI; small intest*; appendi*; colon*; rect*; colorect *; stomach; gastric; duoden*; pancrea*; biliary; bile duct; Vater; ampulla; liver; hepa*]. There was no chronological restriction. Included articles had to have study populations with diagnosed NEN or NEN should be included in the differential diagnosis. They should also have at least 1 ML/DL algorithm for the process of their data, irrespective of the study design. The presence of a comparison group (external validation) was desired but not mandatory. Similarly, the report of at least one benchmarking metric, among accuracy, F1-score, area under receiver operator characteristic curve (AUROC) or area under precision-recall curve (AUPRC) were desired but not mandatory. Table 1 summarizes eligibility criteria. Only full-text publications were considered. Articles not in English language or not providing full text were excluded.

Data extraction was performed by two independent researchers (A.G.P., P.A.P.) using a predefined template with the eligibility and exclusion criteria. In case of disagreement, a third researcher (D.P.L.) made the decision whether to include the article or not. For the collection of relevant data we consulted the Guidelines for Developing and Reporting Machine Learning Predictive Models in Biomedical Research [9]. We collected data on year of publication, country of origin, DOI number, study design (prospective vs. retrospective), classification vs. regression, NEN type studied, dataset (number of patients or samples), input (predictors), output (outcomes), tested AI algorithm(s), training set, test set, internal and external validation sets, cross-validation method, accuracy, F1-score, AUROC (with 95% CI, if available) and AUPRC (with 95% CI, if available).

Numerical variables are presented as mean ± standard deviation (SD). Categorical variables are presented using frequencies and percentages. Calculations and statistical analysis were carried out using the online tool Prism^®^, GraphPad Software, San Diego, CA, USA.

## 3. Results

Literature search across PubMed yielded 1327 articles. In addition, 9 articles were retrieved through other sources (Google^®^ search, screening through articles’ literature). After screening of titles and abstracts, removal of duplicates, and implementation of eligibility criteria, 44 unique articles were included in the final analysis (Figure 1) [10,11,12,13,14,15,16,17,18,19,20,21,22,23,24,25,26,27,28,29,30,31,32,33,34,35,36,37,38,39,40,41,42,43,44,45,46,47,48,49,50,51,52,53].

Regarding geographical distribution (Figure 2), the included studies originated from 12 different countries, with major contributors being the USA (22 studies, 50%), China (12 studies, 27.3%) and Italy (3 studies, 6.8%). Among them, there were 4 coalitions of countries. The studies spanned a 13-year period (2007–2021), with a significant rise over time (Figure 3). Notably, 2/3 of studies were published over 2019–2021, which follows the general increase of publications regarding AI [54].

In order to identify the prediction problem of each study, we collected data on study design, nature of the prediction, and continuity of the target variable, as per Luo et al. [9]. Consequently, there were 19 prospective (42.2%) and 26 retrospective (57.8%) analyses. Notably, one study had 2 stages, one prospective and one retrospective [13], hence the discrepancy between the total number of studies (44) and the sum of analysis based on prospective-retrospective study design (45). Regarding the nature of the prediction, we dichotomized the studies into diagnostic vs. prognostic, depending on whether the prediction referred to healthy subjects or subjects with already diagnosed NET, respectively [55]. The analysis yielded 24 diagnostic (54.5%) and 20 prognostic (45.5%) studies. Finally, all studies but one [24] had to do with classification. The prediction characteristics of each study are summarized in Table 2.

We then classified the papers according to the type of studied NET. Twenty-six studies were about Pan-NETs (59.1%) [10,11,15,16,17,18,19,20,24,25,27,28,30,31,34,38,41,42,43,45,46,47,49,51,52,53], 3 studies had to do with (metastatic) liver NETs (6.8%) [36,37,44], 2 studies analyzed SI-NETs (4.5%) [14,35], whereas colon and rectum [12], rectum [22], non-specified GEP [39], and non-specified GI NETs [50] had from 1 study each (2.3%). There were 4 studies with multiple types of NETs with separate data for each one of them provided (9.1%) [21,23,29,33], and another 2 studies with non-specified multiple types of NETs (4.5%) [13,48]. Figure 4 shows the relevant distribution of studies by NET type.

Regarding the source of data, there were 15 studies with histology-based analyses [10,15,20,23,24,33,38,39,40,41,42,43,45,47,50] and another 15 studies with imaging-based analyses (34.1% each). Six studies were structured based on patient databases (16.7%) [13,22,27,29,32,48], 5 on genetic assays (11.4%) [18,21,30,35,36], and 3 on plasma/serum (6.8%) [12,14,26]. Imaging-based studies were further distinguished in CT-based (6/15, 40%) [17,28,34,46,51,53], EUS-based (4/15, 26.7%) [11,19,25,31], MRI-based (3/15, 20%) [44,49,52], and PET/CT (2/15, 13.3%) [16,37]. Genetic assays included gene expression assays [35,36] and miRNA analyses [18,21] (2 studies each), as well as 1 genome-wide association study (GWAS) [30]. Figure 5 shows the relevant distribution of studies by source of data.

In the set of 44 studies, there were 53 outcome analyses, i.e., 7 studies with more than 1 outcome (5 with two outcomes [13,38,43,45,53], and 2 with three outcomes [15,33]). The most popular outcome analyses were tumor type identification and tumor grade (10 analyses each, 18.9%), tumor detection (5 analyses, 9.4%), and 5-year survival, cell segmentation, disease progression, disease recurrence and Ki-67 scoring (2 analyses each, 3.8%). Table 3 summarizes these outcome analyses, along with the references to relevant studies.

The next analysis we performed was on the number of AI algorithms mentioned within the included studies. As it is expected, a number of studies included more than one AI algorithms, either in an attempt to find the most accurate among them or in the form of comparison of a novel AI model against established ones. In total, we identified 47 different models, with 10 among them being the most utilized ones (Figure 6), i.e., Supporting Vector Classification/Machine (14 analyses, 29.8%) [11,17,21,22,27,28,29,33,36,38,39,47,51,53], Convolutional Neural Network (10 analyses, 21.3%) [15,17,20,23,31,37,40,41,42,49], Random Forest (9 analyses, 19.1%) [11,14,17,22,27,29,46,51,53], Logistic Regression (8 analyses, 17.0%) [11,17,27,29,32,47,51,53], Decision Tree (6 analyses, 12.8%) [11,13,29,33,48,51], Gradient Boosting Decision Tree [29,46,51,53], Multi-Layer Perceptron [25,29,33,47], and (Gaussian) Naïve Bayes [22,29,32,51] (4 analyses each; 8.5%), and AdaBoost [22,46,51], and Linear Discriminant Analysis [10,47,51] with 3 analyses each (6.4%).

We then proceeded with the potential of quantitative assessment of the included studies. Again, we utilized the seminal study of Luo et al. [9] and evaluated the included studies for reporting their training sets, testing sets, cross-validation method and external validation sets. As surrogate metrics of performance for the studied AI algorithms, we considered Accuracy, F1-score, AUROC (95% CI) and AUPRC (95% CI). Only 33 studies out of the included 44 (75%) reported clearly on their training set [10,12,14,15,16,17,19,21,22,23,25,26,27,28,29,30,31,32,33,34,37,39,41,42,43,44,45,46,47,49,50,51,53], 19 mentioned a cross-validation method (43.2%) [10,14,15,17,21,22,25,27,28,29,32,33,34,37,39,45,47,49,51], 36 reported their test set (81.8%) [10,11,13,15,17,18,19,20,21,22,23,25,27,28,29,31,33,34,36,37,38,39,40,41,42,43,44,45,46,47,48,49,50,51,52,53], and only 4 had an external validation set (9.1%) [22,28,42,53]. Thirty-five studies (79.5%) reported at least 1 performance metric in at least 1 dataset (training or test). However, this feature was very heterogenous and non-consistent and we decided not to proceed with further analysis (Appendix A). Regarding training sets, the highest reported Accuracy value was 1.000 (SVM, MLP) [21,33] and the lowest was 0.540 (noisy threshold classifier) [32], the highest reported F1-score was 0.876 (SVC) [29] and the lowest was 0.578 (FCRNA) [43], and the highest reported AUROC was 1.000 (algorithm not specified) [44], while the lowest one was 0.570 (CNN) [17]. With respect to test sets, the highest reported Accuracy value was 1.000 (SVM) [21] and the lowest was 0.310 (CNN) [23], the highest reported F1-score was 0.989 (Decision Tree, Random Forest) [51] and the lowest was 0.578 (FCRNA) [43], and the highest reported AUROC was 1.000 (SVM) [35], whilst the lowest one was 0.462 (Generative Adversarial Domain Adaptation) [45]. Table 2 summarizes the prediction characteristics, the source of data, the implemented AI algorithm(s), and the datasets for each of study included in our scoping review.

## 4. Discussion

This scoping review deals with the current applications of artificial intelligence in the diagnosis and management of gastrointestinal and pancreatic neuroendocrine neoplasms (GEP-NENs). GEP-NENs are inherently rare neoplasms, as such an empirical approach to their management would be unreliable. One of the advantages of AI and its application through machine learning and deep learning is that it can integrate a vast amount of data collected anywhere in the world (big data) and then render them applicable into clinical practice in an individualized manner.

Despite the rarity of NENs, our research yielded a total of 44 relevant studies, the vast majority of which have been published over the last three years. On the one hand, this harmonizes with the general tendency of incremental accumulation of pertinent evidence in Medicine [54,56], on the other hand it may reflect an increasing diagnosis rate of NENs, as it has been documented by the SEER registry [2]. In any case, this establishment may pave the way for future research.

Nevertheless, available studies have several limitations. First, a major restriction are the small datasets of the majority of the studies. There were only 3 among them which used data from large databases with populations of 13,830 [48], 10,580 [22] and 9,663,315 [27] patients, whereas the rest of the studies had populations of 50–361 individuals. Another serious point is that most of the studies did not provide clear information on the structure of the prediction problem (i.e., study design, prognostic vs. diagnostic, classification vs. regression), as such these pieces of information were derived after strenuous digest through the text. Most importantly, there is a non-negligible number of studies with poorly defined training and test sets. Another area of confusion is the lack of universal nomenclature regarding the discrete data sets (i.e., training, validation and test). Some studies use the terms “test set” and “validation set” interchangeably, whereas others are structured based on all three datasets. Future studies should also present their findings on AI algorithm performance in a robust way, including accuracy, F1-score, AUROC and AUPRC, because each one measures different performance aspects and may be a better predictor than the other ones under certain circumstances [57]. Also, such quantification will pave the way for meta-analyses. Furthermore, the ultimate goal of AI is the implementation of the findings of relevant studies into clinical practice. This can be achieved only if the performance of AI algorithms is benchmarked against established tests. Given the small number of studies with an external validation dataset, there is plenty of room for improvement in the field. As mentioned earlier, future endeavors in the field should follow a universal structure as per the existing guidelines, for purposes of both reproducibility and quality [9,58].

As one proceeds from the structure to the content of relevant studies, as we documented, the most popular topics are tumor type identification and grade, tumor detection, 5-year survival, cell segmentation, disease progression, disease recurrence and Ki-67 scoring. In a recent review, Yang et al. showed similar applications of AI with satisfactory prediction accuracy in the diagnosis, risk stratification and prognosis of small intestinal tumors [59]. Interestingly, this review shares 3 studies with the review in hand [14,21,33], which is not surprising given the rarity of small intestinal tumors and the major share of NENs among them. Kim et al. performed a similar analysis of the usefulness of AI in gastric neoplasms [60].

The combination of radiomics, i.e., the multitude of features and technical parameters that can be extracted from imaging studies, with the capability of big data processing offered by AI has opened new frontiers and has led to an exponential burst of pertinent literature. The fundamentals of the process of transforming an imaging study into data that can be processed by an AI algorithm are image acquisition, segmentation (i.e., selection of a region of interest in two dimensions), preprocessing (which allows data homogenization), data extraction, data selection and modelization. Given the routine performance of a constellation of imaging studies in clinical practice, this concept could contribute to the prompt diagnosis of NENs even at a preclinical stage. Promising evidence from imaging of pancreatic tumors with CT and MRI shows that this technology could find more widespread application in the field of NENs [61]. Partouche et al. performed a systematic review and meta-analysis of 161 studies on AI and imaging for Pan-NETs [62]. In accordance with our review, they documented wide heterogeneity of practices, poor procedural compliance with international guidelines, and poor reporting of clinical protocols. They reach the conclusion that standardization and homogenization is the key to future research if AI has the aspiration to enter clinical practice as a standard of care. In an another recent review on the role of radiomics in Pan-NETs, Bezzi et al. also acknowledge the need for further validations before widespread clinical adoption, nevertheless this discipline has great potential in decision-making regarding diagnosis and management [63].

In a process similar to data extraction from imaging studies, histology images can be utilized for processing with the aid of AI algorithms, following a pipeline from whole slide images (WSIs), segmentation into tiles, biomarker visualization and classification. Kuntz et al. recently published a review of 16 studies that used CNN in order to analyze gastrointestinal cancer histology images and showed good performance metrics with external validation, but none of them had clinical implementation for the time being [64].

The main limitation of the review in hand is the heterogeneity of the included studies, on grounds of methodology, dataset allocation and performance benchmarking, which did not allow for a meta-analysis. Structured publications are consequently mandatory in order to facilitate reproducible evidence of high quality. Another predicament for our study is set by the heterogeneity of NENs itself, which may raise methodological limitations. Nevertheless, given the probing nature of our research, an inclusive search strategy was inevitable. Future reviews could focus on specific histologic neuroendocrine types or disease stages.

## 5. Conclusions

To our knowledge, this is the first attempt to systematize existing evidence on the applications of AI in the field of NENs. Published studies focus mostly on diagnosis (tumor detection, tumor identification and tumor grading) rather than management and decision-making, mainly with the use of imaging studies and histology samples. Future directions should take into serious consideration the reporting and quality prerequisites set by already existing guidelines.

## Figures and Tables

**Figure 1 diagnostics-12-00874-f001:**
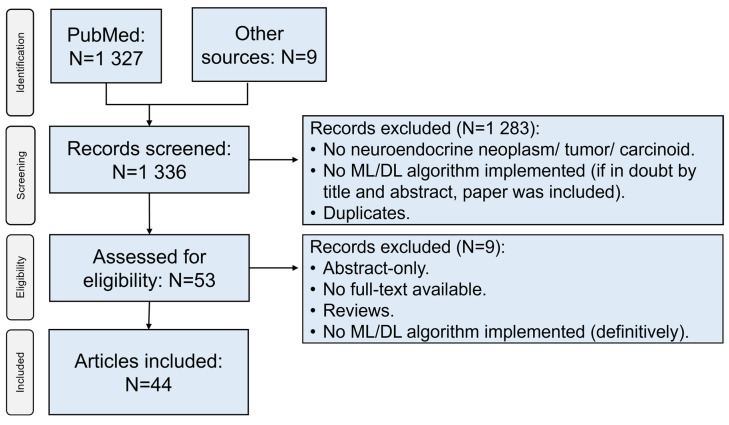
Flowchart depicting the selection process of sources of evidence. ML: machine learning; DL: deep learning.

**Figure 2 diagnostics-12-00874-f002:**
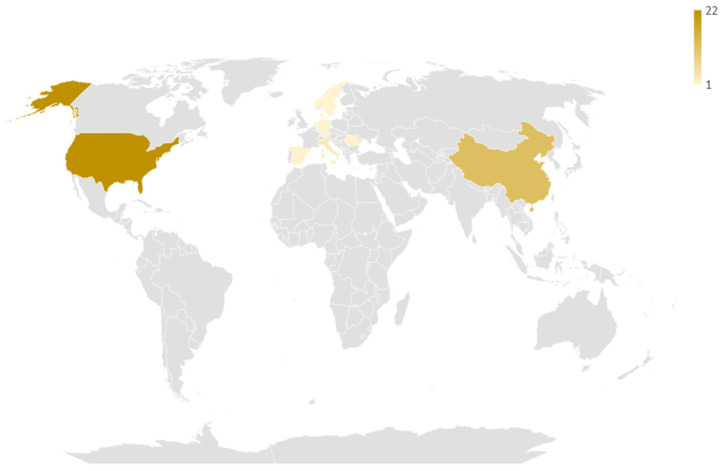
Geographic distribution of the studies included in the review. The darker the hue, the larger the number of studies coming from this particular country.

**Figure 3 diagnostics-12-00874-f003:**
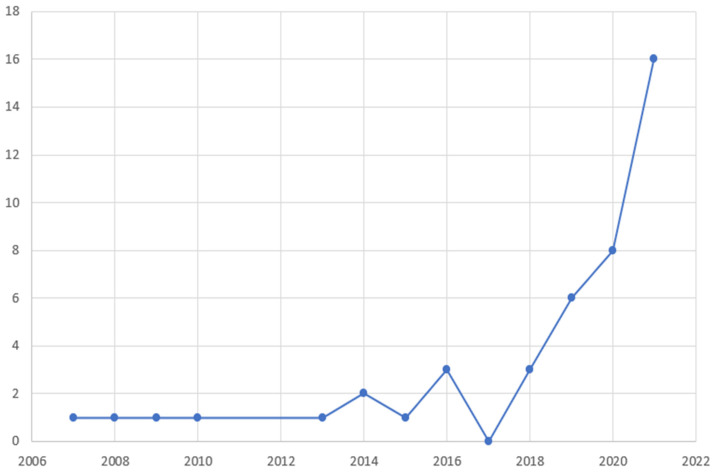
Temporal distribution of the studies included in the review according to year of publication.

**Figure 4 diagnostics-12-00874-f004:**
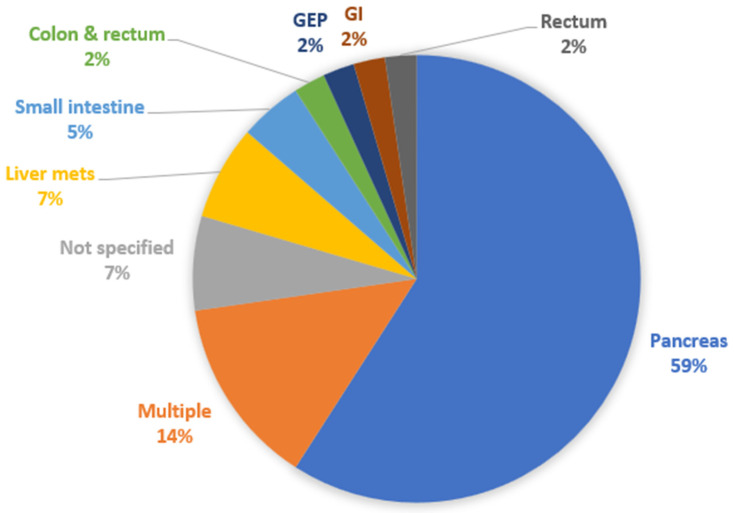
Distribution of studies by type of NET analyzed. GEP: gastroenteropancreatic; GI: gastrointestinal.

**Figure 5 diagnostics-12-00874-f005:**
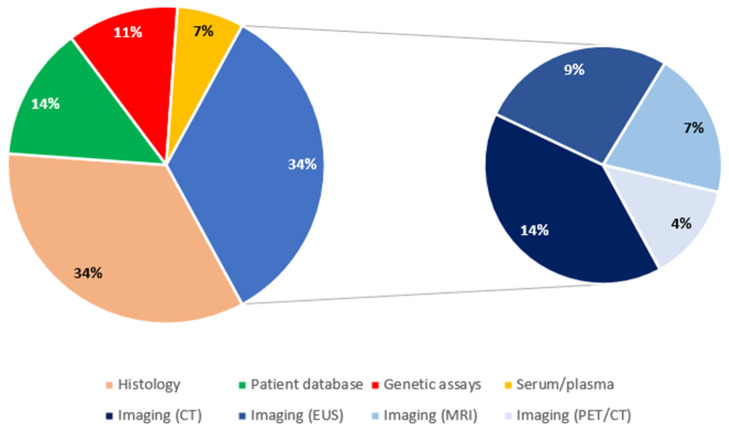
Distribution of studies by source of data. CT: computed tomography; EUS: endoscopic ultrasound; MRI: magnetic resonance imaging; PET: positron emission tomography.

**Figure 6 diagnostics-12-00874-f006:**
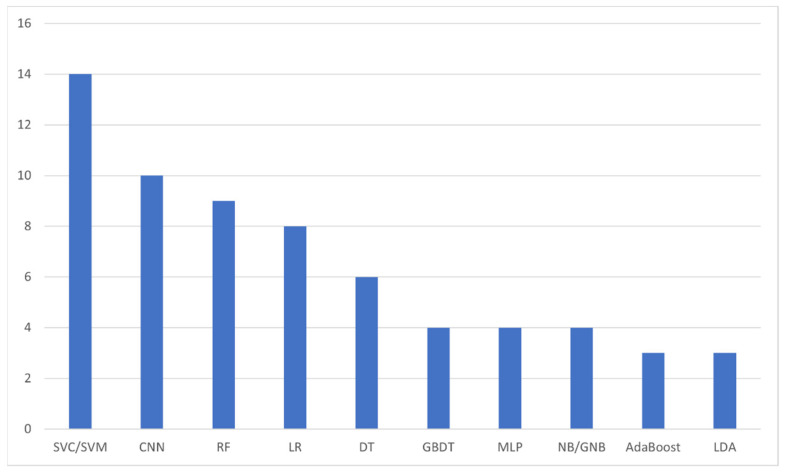
The most frequently appearing artificial intelligence algorithms within the included studies. SVC: Supporting Vector Classification; SVM: Supporting Vector Machine; CNN: Convolutional Neural Network; RF: Random Forest; LR: Logistic Regression; DT: Decision Tree; GBDT: Gradient Boosting Decision Tree; MLP: Multi-Layer Perceptron; NB/GNB: (Gaussian) Naïve Bayes; LDA: Linear Discriminant Analysis.

**Table 1 diagnostics-12-00874-t001:** Inclusion criteria.

Parameter	Inclusion Criteria
Population	Diagnosed cases with NEN (NET/NEC) or NEN included in the differential diagnosis.
Intervention	Analysis with a ML/DL algorithm.
Comparison	External validation desired but not mandatory.
Outcome	Report of accuracy, F1-score, AUROC or AUPRC desired but not mandatory.
Study design	Any. Abstract-only studies were excluded

NEN: neuroendocrine neoplasm; NET: neuroendocrine tumor; NEC: neuroendocrine carcinoma; ML: machine learning; DL: deep learning; AUROC: area under receiver operator characteristic (ROC) curve; AUPRC: area under precision-recall (PR) curve.

**Table 2 diagnostics-12-00874-t002:** Collective representation of the studies included in the present review, with respective prediction characteristics, technical characteristics, datasets and benchmarking. For reasons of conciseness, we have included only AUC of all the mentioned benchmarking measurements.

Study ID				Prediction Characteristics			Technical Characteristics			Datasets & Benchmarking						
First Author	Year of Publication	DOI	Ref. No.	Study Design	Nature of Prediction	Continuity of Output	NET Type	Source of Data	Tested AI Algortihm(s)	Training	AUC-Training	Cross-Validation	Test	AUC-Test	Ext. Validation	AUC
Bevilacqua A	2021	10.3390/diagnostics11050870	[10]	Prospective	Prognostic	Classification	Pancreas	Histology	LDA-model A	Y	0.870–0.940	3-fold x100	Y	0.870–0.900	N	
Chen K	2018	10.1016/S1470-2045(20)30323-5	[11]	Retrospective	Prognostic	Classification	Pancreas	Imaging (EUS)	DT, LR, NN, RF, SVM	N		N	Y	0.879–0.997	N	
Cheng X	2021	10.3389/fsurg.2021.745220	[22]	Retrospective	Prognostic	Classification	Rectum	Database	AdaBoost, NB, Nu-SVC, SVC, RF, XGB	Y	0.780–0.850	10-fold	Y	0.890	Y	0.830–0.890
Drozdov I	2009	10.1002/cncr.24180	[33]	Prospective	Diagnostic	Classification	Primary small intestine; metastatic liver	Histology	DT, SVM	Y		10-fold	Y		N	
Drozdov I	2009	10.1002/cncr.24180	[33]	Prospective	Prognostic	Classification	Primary small intestine; metastatic liver	Histology	Perceptron	Y		N	N		N	
Fehrenbach U	2021	10.3390/cancers13112726	[44]	Prospective	Prognostic	Classification	Liver	Imaging (MRI)	Not specified	Y	0.908–1.000	N	Y		N	
Gao X	2019	10.1007/s11548-019-02070-5	[49]	Prospective	Prognostic	Classification	Pancreas	Imaging (MRI)	CNN	Y	0.915 *	5-fold	Y	0.893 *	N	
Govind D	2020	10.1038/s41598-020-67880-z	[50]	Prospective	Prognostic	Classification	GI	Histology	deep-SKIE, SKIE (GAN-based), deep-SKIE (GAN-based)	Y		N	Y		N	
Han X	2021	10.3389/fonc.2021.606677	[51]	Retrospective	Diagnostic	Classification	Pancreas	Imaging (CT)	AdaBoost, DT, GBDT, GNB, KNN, LDA, LR, SVM, RF	Y		10-fold x1000	Y	0.946–0.997 *	N	
Huang B	2021	10.1109/JBHI.2020.3043236	[52]	Retrospective	Prognostic	Classification	Pancreas	Imaging (MRI)	DFSR	N		N	Y	0.919	Y	0.688–0.840
Huang B	2021	10.1109/JBHI.2021.3070708	[53]	Retrospective	Prognostic	Classification	Pancreas	Imaging (CT)	GBDT, LR, RF, SVM	Y	0.660–0.760	N	Y	0.700–0.870	Y	0.710–0.830
Ito H	2020	10.4251/wjgo.v12.i11.1311	[12]	Retrospective	Diagnostic	Classification	Colon & rectum	Serum	BT	Y		N	N		N	
Kidd M	2021	10.1159/000508573	[13]	Retrospective	Prognostic	Classification	Multiple	Database		N		N	N		N	
Kidd M	2021	10.1159/000508573	[13]	Prospective	Prognostic	Classification	Multiple	Database	DT	N		N	Y		N	
Kjellman	2021	10.1159/000510483: 10.1159/000510483	[14]	Prospective	Diagnostic	Classification	Small intestine	Serum	RF	Y	0.970–0.990	5-fold	N		N	
Klimov S	2021	10.3389/fonc.2020.593211	[15]	Retrospective	Diagnostic	Classification	Pancreas	Histology	CNN	Y		5-fold	Y		N	
Klimov S	2021	10.3389/fonc.2020.593211	[15]	Retrospective	Prognostic	Classification	Pancreas	Histology	CNN, ML “zoo” (18 different models)	Y		5-fold, leave-one-out	N		N	
Liu Y	2014	10.1016/j.media.2014.02.005.	[16]	Prospective	Prognostic	Classification	Pancreas	Imaging (PET/CT)	RDM	N		N	N		N	
Luo Y	2019	10.1159/000503291	[17]	Retrospective	Prognostic	Classification	Pancreas	Imaging (CT)	CNN, LR, RF, SVM	Y	0.570–0.810	8-fold	Y	0.820	N	
Nanayakkara J	2020	10.1093/narcan/zcaa009	[18]	Retrospective	Diagnostic	Classification	Pancreas	miRNA	data mining	N		N	Y		N	
Nguyen VX	2010	10.7863/jum.2010.29.9.1345	[19]	Retrospective	Diagnostic	Classification	Pancreas	Imaging (EUS)	ANN	Y		N	Y	0.890	N	
Niazi MKK	2018	10.1371/journal.pone.0195621	[20]	Retrospective	Diagnostic	Classification	Pancreas	Histology	Inception v3-C1 (type of CNN), Bootstrapped Inception v3-C1	N		N	Y	0.922–0.973	N	
Panarelli N	2019	10.1530/ERC-18-0244	[21]	Retrospective	Diagnostic	Classification	Appendix, GEP, ileum, pancreas, rectum	miRNA	SVM	Y		10-fold	Y		N	
Redemann J	2020	10.4103/jpi.jpi_37_20	[23]	Retrospective	Diagnostic	Classification	Appendix, colon & rectum, duodenum, pancreas, small intestine, stomach, total (icl. lung)	Histology	CNN	Y		N	Y		N	
Saccomandi P	2016	10.1007/s10103-016-1948-1	[24]	Retrospective	Prognostic	Regression	Pancreas	Histology	Inverse Monte Carlo	N		N	N		N	
Saftoiu A	2008	10.1016/j.gie.2008.04.031	[25]	Prospective	Diagnostic	Classification	Pancreas	Imaging (EUS)	MLP	Y		10-fold	Y		N	
Soldevilla B	2021	10.3390/cancers13112634	[26]	Prospective	Diagnostic	Classification	Not specified	Plasma	OPLS-DA supervised model	Y	0.779–0.982	N	N		N	
Song Y	2018	10.7150/jca.26649	[27]	Retrospective	Prognostic	Classification	Pancreas	Database	DL, LR, SVM, RF	Y		10-fold	Y	0.870 (DL)	N	
Song C	2021	10.21037/atm-21-25	[28]	Retrospective	Prognostic	Classification	Pancreas	Imaging (CT)	SVM (various models)	Y	0.580–0.830	10-fold	Y	0.480–0.770	Y	0.520–0.560
Telalovic JH	2021	10.3390/diagnostics11050804	[29]	Retrospective	Prognostic	Classification	GI; pancreas	Database	DT, GB GNB, KNN, MLP, MNB, LR, RF, SVC, XT	Y		10-fold	Y		N	
Tirosh A	2019	10.1002/cncr.31930	[30]	Prospective	Diagnostic	Classification	Pancreas	GWAS	Unsupervised clustering analysis	N		N	N		N	
Udristoiu AL	2021	10.1371/journal.pone.0251701	[31]	Prospective	Diagnostic	Classification	Pancreas	Imaging (EUS)	CNN-LSTM (different models)	Y		N	Y	0.970–0.990	N	
van Gerven MAJ	2007	10.1016/j.artmed.2006.09.003	[32]	Retrospective	Prognostic	Classification	Not specified	Database	NTC	Y		leave-one-out	N		N	
Wan Y	2021	10.1002/mp.15199	[34]	Retrospective	Prognostic	Classification	Pancreas	Imaging (CT)	SAE, hybrid (SAE+handcrafted)	Y	0.766–0.934	5-fold	Y	0.739	N	
Wang Q	2020	10.1042/BSR20193860	[35]	Prospective	Diagnostic	Classification	Small intestine	Gene expression assay	ANN	N		N	N		N	
Wang Q	2021	10.3389/fonc.2021.725988	[36]	Retrospective	Diagnostic	Classification	Liver	Gene expression assay	SVM	N		N	Y	0.945–1.000	N	
Wehrend J	2021	10.1186/s13550-021-00839-x	[37]	Retrospective	Diagnostic	Classification	Liver	Imaging (PET/CT)	CNN	Y		5-fold	Y	0.700–0.730 **	N	
Xing F	2013	10.1007/978-3-642-40811-3_55	[38]	Prospective	Diagnostic	Classification	Pancreas	Histology	SVM	N		N	Y		N	
Xing F	2014	10.1109/TBME.2013.2291703	[39]	Prospective	Diagnostic	Classification	GEP	Histology	SVM	N		3-fold	N		N	
Xing F	2015	10.1007/978-3-319-24574-4_40	[40]	Prospective	Diagnostic	Classification	Not specified	Histology	CNN	N		N	Y		N	
Xing F	2016	10.1007/978-3-319-46726-9_22	[41]	Prospective	Diagnostic	Classification	Pancreas	Histology	CNN	Y		N	Y		N	
Xing F	2016	10.1109/TMI.2015.2481436	[42]	Prospective	Diagnostic	Classification	Pancreas	Histology	CNN	Y		N	Y		N	
Xing F	2019	10.1109/TBME.2019.2900378	[43]	Prospective	Diagnostic	Classification	Pancreas	Histology	FCN-8s, FCRNA, FCRNB, FRCN, KiNet, SFCNOPI, U-Net	Y		N	Y	0.525–0.724	N	
Zhang X	2020	10.1200/CCI.19.00108	[45]	Retrospective	Diagnostic	Classification	Pancreas	Histology	GADA	Y	0.627–0.857	2-fold	Y	0.462–0.775	N	
Zhang T	2021	10.3389/fonc.2020.521831	[46]	Retrospective	Prognostic	Classification	Pancreas	Imaging (CT)	DC + AdaBoost, DC + GBDT, XGB + RF	Y		N	Y	0.570–0.860	N	
Zhou RQ	2019	10.12998/wjcc.v7.i13.1611	[47]	Retrospective	Prognostic	Classification	Pancreas	Histology	LDA, LR, MLP, SVM	N		leave-one-out	Y		N	
Zimmerman NM	2021	10.2217/fon-2020-1254	[48]	Retrospective	Prognostic	Classification	Multiple	Database	DT	N		N	N		N	

* Only the algorithm with the best performance is mentioned. ** AUPRC (instead of AUROC).

**Table 3 diagnostics-12-00874-t003:** Most popular outcome analyses within the included studies.

Outcome	Number of Studies (%)	Reference No.
Tumor type identification	10 (18.9)	[12,18,19,21,23,25,31,36,37,51]
Tumor grade	10 (18.9)	[10,11,17,34,46,47,49,50,52,53]
Tumor detection	5 (9.4)	[14,20,26,33,43]
5-year survival	2 (3.8)	[22,27]
Cell segmentation	2 (3.8)	[40,42]
Disease progression	2 (3.8)	[13,29]
Disease recurrence	2 (3.8)	[28,53]
Ki-67 scoring	2 (3.8)	[38,39]

## Data Availability

Not applicable.

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
