# Peer review of "Artificial Intelligence and Machine Learning in the Diagnosis and Management of Gastroenteropancreatic Neuroendocrine Neoplasms—A Scoping Review"

_diagnostics, 2022, doi:10.3390/diagnostics12040874_

Round 1

Reviewer 1 Report

This is a manuscript on AI use in GEP NENs.

I have some minor comments:

Line 41: The functionality of the tumors is not depending on the secretion of the hormones because it can be that the hormones that are secreted are inactive. The functionality depends in the fact that the hormones give rise to a syndrome.

Lines 45 and 46: Gastric NETs can give rise to atypical carcinoid syndrome and has to be included.

My main concern is that the present review has included a variety of NENs (methodology issues) that differ from each other and thus the results were negative concluding that AI can not be used yet in clinical praxis.

Author Response

Dear reviewer,

Thank you very much for your time and your insightful comments.  We hereby cite our responses to your comments.

1) Line 41 (on functionality of NETs): We have removed the phrase "depending whether they secrete hormones or not".

2) Lines 45-46 (on gastric NETs): We have added a concise passage on gastric NETS and the characteristics of each type at the end of paragraph 1 of the Introduction.

3) We have addressed the potential methodological issues stemming from the variety of NENs in the section of limitations within the last paragraph of the Discussion.

Once again, thank you for your review and your constructive feedback.

Reviewer 2 Report

This is a clear and informative review about the use of artificial intelligence, machine learning and deep learning in the context of gastroenteropancreatic neuroendocrine neoplasms.

The methodology adopted isprecise and well described. Results are also clear and informative.

Please find here below few aspects that need to be clarified/improved.

RESULTS

-please check the legend of Figure 3

- Whenever statistics or data is reported, all the corresponding publications should be cited (as done in lines 217-225, reference should be reported thoruthough all the manuscript).  For instance, when authors state “Only 28 studies out of the included 44 (63.6%) reported clearly on their training set, 20 mentioned a cross- validation method (45.5%), 36 reported their test set (81.8%), and only 4 had an external validation set (9.1%).”, references should be reported so that the reader can be able to retrieve those works.

The same for “In total, we identified 195 48 different models, with 11 among them being the most utilized ones (Figure 6), i.e. Sup- 196 porting Vector Classification/Machine (12 analyses, 25%), Convolutional Neural Network 197 and Random Forest (9 analyses each, 18.8%), Logistic Regression (8 analyses, 16.7%), Decision Tree (6 analyses, 12.5%), Gradient Boosting Decision Tree and Multi-Layer Perceptron (4 analyses each; 8.3%), and AdaBoost, Distance Correlation, Linear Discriminant 200 Analysis and Naïve Bayes, with 3 analyses each (6.25%).”… it is now not possible to retrieve the works using CNNs. Please carefully revise all the manuscript and provide the citations.

- It would be helpful to have a summarizing table reporting  with the most relevant characteristics of the 44 included articles. Several interesting features have been explored in the present manuscript, such as numbers and percentages for each study; to summarize all these characteristics in a comprehensive table would help the reader to gain an overall view of the state-of-the-art. Moreover, it would be much easier to select among the 44 revised articles, the ones of interest for the reader, based on the desired characteristics. 

In addition, among lines 233-278 I would suggest to add the following review : Bezzi C et al.

Radiomics in pancreatic neuroendocrine tumors: methodological issues and clinical significance. Eur J Nucl Med Mol Imaging. 2021, Nov;48(12):4002-4015.

Author Response

Dear reviewer,

Thank you for your constructive critique on our work.  Your comments were important and very helpful in that we spotted some errors in our previous manuscript.  More specifically:

1) We have changed the legend of Figure 3 (it was a copy-paste type of error).

2) We revised the entire manuscript and provided references wherever they were absent.

3) After scrutinizing the included references, we revised the most frequently appearing AI algorithms, we made the respective changes in the frequencies within the text and we redesigned the relevant graph (Figure 6).

4) We added a concise, but collective and informative table (Table 3) which summarizes all the prediction characteristics, technical aspects, and datasets & benchmarking of the included studies.

5) We incorporated the suggested review on radiomics in PanNETs by Bezzi et al. at the end of paragraph 5 of the Discussion.

Once again, on behalf of the authors, I would like to express our gratitude for your thorough perusal through our manuscript.